# Iodine Biofortification of Dandelion Plants (*Taraxacum officinale* F.H. Wiggers Coll.) with the Use of Inorganic and Organic Iodine Compounds

**DOI:** 10.3390/molecules28155638

**Published:** 2023-07-25

**Authors:** Iwona Ledwożyw-Smoleń, Joanna Pitala, Sylwester Smoleń, Marta Liszka-Skoczylas, Peter Kováčik

**Affiliations:** 1Faculty of Biotechnology and Horticulture, University of Agriculture in Kraków, Al. Mickiewicza 21, 31-120 Kraków, Poland; sylwester.smolen@urk.edu.pl; 2Laboratory of Mass Spectrometry, University of Agriculture in Kraków, Al. Mickiewicza 21, 31-120 Kraków, Poland; joanna.pitala@urk.edu.pl; 3Department of Engineering and Machinery for Food Industry, Faculty of Food Technology, University of Agriculture in Krakow, Al. Mickiewicza 21, 31-120 Kraków, Poland; marta.liszka-skoczylas@urk.edu.pl; 4Department of Agrochemistry and Plant Nutrition, Institute of Agronomic Sciences, Faculty of Agrobiology and Food Resources, Slovak University of Agriculture in Nitra, Tr. A. Hlinku 2, 949 01 Nitra, Slovakia; peter.kovacik@uniag.sk

**Keywords:** iodine, biofortification, iodosalicylates, organic iodine compounds, dandelion, antioxidant potential

## Abstract

Iodine is a crucial microelement necessary for the proper functioning of human and animal organisms. Plant biofortification has been proposed as a method of improving the iodine status of the population. Recent studies in that field have revealed that iodine may also act as a beneficial element for higher plants. The aim of the work was to evaluate the efficiency of the uptake and accumulation of iodine in the plants of dandelion grown in a pot experiment. During cultivation, iodine was applied through fertigation in inorganic (KI, KIO_3_) and organic forms (5-iodosalicylic acid, 5-ISA; 3,5-diiodosalicylic acid, 3,5-diISA) at two concentrations (10 and 50 µM). The contents of total iodine and iodosalicylic acids, as well the plant biomass and antioxidant capacity of dandelion leaves and roots, were analyzed. The uptake of inorganic and organic forms by dandelion plants was confirmed with no negative effect on plant growth. The highest efficiency of improving iodine content in dandelion leaves and roots was noted for 50 µM KI. The applicability of iodosalicylates, especially 5-ISA, for plant biofortification purposes was confirmed, particularly as the increase in the iodine content after the application of 5-ISA was higher as compared to that with commonly used KIO_3_. The chemical analyses have revealed that iodosalicylates are endogenous compounds of dandelion plants.

## 1. Introduction

Iodine is a crucial element for the proper functioning of human and animal organisms. Its necessity results from its presence in thyroid hormones: triiodothyronine and thyroxine. The recommended daily intake of iodine has been established on the following levels: 90 µg I for 0–6 yo children, 120 µg I for 6–12 yo children, 150 µg for adults and 250 µg I for pregnant and lactating women [1]. The insufficient supply of iodine may lead to, among others, malfunction of the thyroid gland, increased risk of pregnancy loss, growth retardation and neurological deficits grouped as IDDs—iodine deficiency disorders [1]. The main sources of iodine in a daily diet include: table salt, dairy products, bread, meat and fish [2]. In the coastal areas, iodine deficiency is less likely to occur due to the substantial consumption of marine fish that are naturally rich in that element. In the terrestrial regions, however, the problem of the insufficient supply of iodine needs to be monitored and counteracted. One of the major factors contributing to the increased risk of iodine deficiency in the human population is the low level of that element in plant food products, which results from the low accumulation and mobility of iodine in the soil. It has been established that major factors affecting iodine bioavailability in the soil include the soil content of organic matter, Fe and Al oxides, soil texture, pH and Eh [3]. Iodine sorption in the soil occurs by binding with soil organic matter (SOM), as well as hydrous aluminum and iron oxides, the latter case being particularly important for iodates when soil pH < 6 [4]. The sorption of iodides occurs more rapidly than that of iodates [4]. Prior to the process, iodate is reduced and iodide is oxidized to more reactive species, such as I_2_ or HIO, which can then be incorporated into the SOM [5,6]. Iodine release from the soil depends strongly on the soil redox potential and pH. Therefore, in soils rich in organic matter, the amount of exchangeable, and therefore available iodine to plants, is limited [6]. It has been stated that only a small percentage of the total iodine in the soil is present as water-soluble and therefore a highly available form for plants [3].

Most likely, iodide ion uptake by plants occurs directly through chloride ion channels [7]. The uptake of iodate requires its previous reduction to an iodide form [8]. The exact mechanism and the responsible enzyme are still to be determined, yet the possible involvement of nitrate reductase has been previously suggested [9]. The requirement of iodate reduction prior to uptake may be a major factor underlying the observed more effective uptake of iodides over iodates. Recent research has indicated a possible favoring effect of low iodine doses on various processes in higher plants, including growth, the uptake of selected mineral nutrients and increases in antioxidant potential; for a review see [10,11,12]. The mechanism of the biostimulative action of iodine on higher plants is not yet recognized; still, the presence of various iodinated proteins related, among others, to photosynthetic activity, has recently been revealed in *Arabidopsis thaliana* [11]. These findings allowed for a proposal to qualify iodine into the group of beneficial elements for plants [13,14]. However, excessive iodine doses can impose toxic effects, the occurrence of which depends on the plant genotype, dose and method of cultivation [12]. At the same time, there is no information regarding the possible interaction of exogenous iodine with plant hormones. Phytohormones involved in plant reactions to both biotic and abiotic stress include jasmonic acid (JA) and salicylic acid (SA) [15].

Plant biofortification is an agronomic or biotechnological approach that aims at increasing the content of necessary nutrients, such as iron, calcium, zinc and selenium in the crop [16]. So far, most studies conducted on iodine enrichment in higher plants used inorganic iodine compounds, such as iodides and iodates, as the source of that element. However, the possible root uptake of organic forms of iodine by higher plants and its further translocation and accumulation have recently been more intensely studied providing some promising results, in terms of the iodine biofortification process. It has been revealed that among tested exogenous organoiodine compounds, those being derivatives of salicylic acid, such as 5-iodosalicylic acid (5-ISA) and 3,5-diiodosalicylic acid (3,5-diISA), are efficiently taken up and pose no exceptional toxicity towards plants [17,18,19]. What is more, 5-ISA and 3,5-diISA increased the health-promoting properties of lettuce plants, as demonstrated in in vitro studies on human cancer cell lines [20]. It needs to be mentioned that iodosalicylates have recently been reported to occur as endogenous compounds in higher plants, i.e., lettuce [18].

The possibility of targeting herbal and medicinal plants for biofortification purposes has been studied only for the last few years. However, they seem to be of great interest, not only from a human health perspective but also regarding the supplementation of deficient minerals in fodder. Some research was performed on basil [21,22,23,24] and garlic [25,26,27], with selenium being the most studied microelement. Some trials have been undertaken regarding the possibility of improving the iodine content in medicinal plants [21,28,29]. Dandelion (*Taraxacum officinale* F.H. Wiggers Coll.) is a perennial plant belonging to the *Asteraceae* family. It is widely popular in Europe, Asia and North America. It is one of the most common medicinal plants, particularly as almost all of its parts exhibit substantial pro-health, including antioxidant, anti-inflammatory, hypoglycemic and hepatoprotective properties [30,31]. Despite its popularity, no studies on the mineral biofortification of dandelion have been conducted so far.

The primary aim of the work was to evaluate the possibility of improving iodine content in dandelion plants through the root application of different doses of inorganic (KI and KIO_3_) and organic (5-iodosalicylic acid, 5-ISA; 3,5-diiodosalicylic acid, 3,5-diISA) iodine forms. It was also aimed at describing the effect of iodosalicylates on plant growth and the content of selected antioxidative compounds that may act as stress indicators in plants.

## 2. Results

### 2.1. Growth of Dandelion Plants

The application of tested inorganic and organic iodine compounds had no significant effect on the growth of dandelion plants, as measured by the weights of leaves and roots of a single plant (Table 1). The dry matter content in leaves and roots was however affected. As compared to the control, an increase in dry matter content in leaves was noted for the combination of KI 50 µM and 3,5-diISA. The lowest values of that parameter were found in leaves of plants grown in the presence of KIO_3_ at both applied doses (10 and 50 µM). As compared to the control combination, dry matter content in roots decreased significantly in plants with combinations of 50 µM doses of KI, 5-ISA and 3,5-diISA.

### 2.2. Iodine Content in Dandelion Roots and Leaves

Plant cultivation in the substrate enriched with tested inorganic and organic iodine compounds increased the total content of iodine in leaves and roots of dandelion plants (Table 2). The highest total accumulation of that element in dandelion plants was found for the application of 50 µM KI: 133.5 mg I kg^−1^ and 19.1 mg I kg^−1^ in leaves and roots, respectively. A substantial level of iodine uptake was also noted after the application of a 50 µM I dose of 5-ISA (99.2 and 17.1 mg I kg^−1^ in leaves and roots, respectively). Between the two inorganic sources of iodine, namely KI and KIO_3_, a higher effect of iodine enrichment in both roots and leaves was noted for KI irrespective of the applied dose. In the case of organic sources of iodine, the significant differentiation between 5-ISA and 3,5-diISA, in terms of increasing the total I content in leaves and roots, was observed only for the higher dose (50 µM). The application of 50 µM 5-ISA led to a greater accumulation of total iodine in both plant parts (99.2 and 17.1 mg I kg^−1^ in leaves and roots, respectively) as compared to that with the same concentration of 3,5-diISA (38.5 and 9.1 mg I kg^−1^ in leaves and roots, respectively). The application of 50 µM 5-ISA, an organic source of iodine, allowed to obtain a higher level of iodine in dandelion plants (both leaves and roots), compared to the inorganic compound KIO_3_ applied at the same dose.

The contents of 5-ISA and 3,5-diISA were analyzed in the leaves and roots of dandelion plants in all tested combinations (Table 2). There was little differentiation in the content of 5-ISA in dandelion leaves, and a significant increase in its level was noted only after the application of 50 µM 5-ISA. On the other hand, the content of 5-ISA in roots was not different from the control value in the case of both doses of KIO_3_ and 10 µM KI. The highest level of that compound (6.3 ng g^−1^) was noted in the roots of plants grown in the substrate with 50 µM I 5-ISA. The applied dose of 3,5-diISA did not affect the level of 5-ISA in dandelion roots. Additionally, increasing the KI dose from 10 to 50 µM I led to a significantly higher content of 5-ISA in roots (0.1 and 0.8 ng g^−1^, respectively). A significant increase in the leaf level of 3,5-diISA was noted only for the combinations with 3,5-diISA application, with the highest level noted for the 50 µM concentration: 8.88 ng 3,5-diISA g^−1^. In the case of roots, an increase in the 3,5-diISA level was noted for both doses of 3,5-diISA, as well as the 50 µM dose of 5-ISA. In general, the content of both iodosalicylates was higher in roots than in leaves of dandelion plants. Analyses of control plants showed that iodosalicylates are endogenous compounds of dandelion plants.

### 2.3. The Content of Phenolic Compounds and Antioxidant Capacity of Dandelion Plants

The application of inorganic and organic sources of iodine modified the level of phenolic compounds in dandelion leaves and roots (Table 3 and Table 4). The highest total content of phenolics was noted in the leaves of plants grown in the presence of 50 µM KIO_3_ (50.2 mg CAE g^−1^), while the lowest was in the leaves of plants from the combinations: 10 µM KI (37.2 mg CAE g^−1^) and 50 µM 5-ISA (36.5 mg CAE g^−1^). In all remaining combinations, the leaf accumulation of phenolic compounds did not differ significantly from the value noted for the control plants. The root content of total phenolics was the highest for the combinations with KI application (10.2 and 11.7 mg CAE g^−1^, for 10 and 50 µM I, respectively). Increased root levels of these compounds, as compared to the control, were also noted for 50 µM doses of 5-ISA (9.0 mg CAE g^−1^) and 3,5-diISA (9.7 mg CAE g^−1^).

For most tested treatments, the introduction of iodine compounds into the substrate decreased the level of salicylic acid in plants, as compared to the control (Table 3 and Table 4). Only in the case of 50 µM KI was an increase in the SA content in leaves observed (69.0 ng g^−1^, Table 3). The increase in the applied iodine dose had no effect on the SA level in dandelion leaves, with the exception of higher SA content after the application of 50 µM KI, as compared to that with 10 µM KI. In the case of roots, the iodine dose variously affected SA accumulation depending on the tested compound (Table 4). For 50 µM KI and 3,5-diISA, there was a noted decrease, while for KIO_3_, there was an increase in the SA level as compared to respective combinations with lower dose of iodine. The applied dose of 5-ISA had no effect on SA accumulation in dandelion roots.

There was significant differentiation between the tested combinations with respect to the accumulation of jasmonic acid (JA) in dandelion leaves and roots (Table 3 and Table 4). The introduction of inorganic and organic iodine sources into the substrate increased the content of JA in dandelion leaves, as compared to that in the control plants (Table 3). The highest level of that compound was noted for treatments with both doses of 3,5-diISA (182.4 and 175.2 ng g^−1^), as well as the 50 µM dose of 5-ISA (164.6 ng g^−1^). The applied iodine dose had no significant effect on the leaf level of JA from KIO_3_, KI and 3,5-diISA treatments. Only in the case of 5-ISA application did an increase in the iodine dose lead to an increase in the JA level in dandelion leaves. The content of JA in dandelion roots decreased significantly, as compared to the control, after the application of KI, 5-ISA and 3,5-diISA (Table 4). Only the introduction of KIO_3_ into the substrate led to a significantly increased level of JA in roots (432.0 and 439.3 ng g^−1^). Increasing the dose of iodosalicylates (5-ISA and 3,5-diISA) from 10 to 50 µM decreased the accumulation of JA in roots; such an effect was not noted for KI.

There was little differentiation observed in the antioxidant capacity of dandelion leaf extracts, as measured using the CUPRAC method (Table 5). The highest CUPRAC values were noted in the leaves of plants fertigated with 3,5-diISA in both doses (214.6 and 215.1 µmol TE g^−1^). However, when individual combinations were taken into account, only in the case of 50 µM KI and 3,5-diISA applications did the obtained values differ significantly from the control. FRAP values of dandelion leaves increased significantly and similarly with almost all tested combinations, as compared to the control values, with the exception of 10 µM KIO_3_ and 50 µM KI (Table 5). Additionally, the FRAP value for leaves with the 10 µM KIO_3_ combination was significantly lower than those obtained when 5-ISA and 3,5-diISA were applied at both doses, as well as KI applied at a 10 µM KI dose. Little effect of iodine compounds was found with respect to the radical scavenging capacity of leaf extracts, as measured with the use of the DPPH method (Table 5). A significant increase in that parameter, as compared to the control, was noted only in plants fertigated with 50 µM KIO_3_ (72.8 µmol TE g^−1^).

The antioxidant capacity of dandelion roots, as measured using CUPRAC, FRAP and DPPH methods, increased significantly, as compared to the control, in combination with KI application at both doses (Table 6). The lowest levels of FRAP values were noted in plants fertigated with 10 µM 3,5-diISA. An increase in the applied dose from 10 to 50 µM I led to a significant increase in the antioxidant potential, as measured by all used methods, in the roots of plants fertigated with 3,5-diISA. An increase in the KIO_3_ dose contributed to significantly higher levels of CUPRAC and FRAP values of dandelion roots, while in the case of KI, a significant increase in the FRAP value was accompanied by the increased I dose. The application of a higher iodine dose of 5-ISA decreased the CUPRAC value of dandelion roots with no effect on the antioxidant potential, as measured via FRAP and DPPH methods.

## 3. Discussion

For years, iodine was not considered an important mineral element for plants. Only limited information was available considering the potential beneficial effect of that element on crop plant growth, while practical applications were limited to being a defoliating and herbicide agent [32]. In the last decade, however, an interest in that element has increased, in large part owing to the possibility of improving the human intake of iodine through the consumption of the so-called biofortified food. The on-going research on the potential of increasing the iodine content in edible plants has substantially improved the recognition of the role of iodine in higher plants. It has been discovered that relatively low doses of iodine not only can be neutral (non-toxic) for plants but also may increase their growth (for a review see [10,12]). The results obtained in the present study showed no significant effect of inorganic and organic sources of iodine, applied at two different but still low doses (10 µM and 50 µM I), with respect to the growth of dandelion plants in pot cultivation. This stays in general agreement with other studies reporting the effect of similar iodine doses during plant cultivation. No toxic effects on lettuce growth were noted when iodine (KI or KIO_3_) was applied in the hydroponics nutrient solution in the 0–80 µM I concentration range [33]. The studies by Kiferle et al. [28] have revealed that KI and KIO_3_ applied at a 100 µM I dose had no effect on the dry weight of basil plants grown in the soil. However, there is limited information on safe doses of iodine applied as organic compounds. Grzanka et al. [34] noticed an increase in the whole plant and root system growth of young corn plants after the application of 10 µM I of 5-ISA and 2-IBeA (2-iodobenzoic acid). Smoleń et al. [18] found that 5-ISA and 3,5-diISA, at the concentrations of 10 and 20 µM I, had no effect on the fresh weight of lettuce grown in mineral or peat soils. Other work has reported, however, no negative effect of 5-ISA, at a concentration up to 8 µM I, on the biomass of lettuce grown in the hydroponic system, while above that threshold, strong deformation of leaves and roots was observed [35]. Quite similarly, in another study with the hydroponic cultivation of lettuce, the unfavorable effect of iodosalicylates (5-ISA and 3,5-diISA) on plant growth was noted for concentrations as low as 10 µM [36]. These observations indicate a substantial impact of the method of plant cultivation on the potential toxicity of applied iodosalicylates. However, so far, no studies comparing the effect of the cultivation method on the toxicity levels of various organic iodine compounds have been conducted.

The use of organic iodine compounds as a source of that element for plant enrichment purposes is a relatively new direction in crop biofortification. Most of the so-far conducted studies were focused on the possible application of inorganic compounds, namely iodides (I^−^) and iodates (IO_3_^−^, [10]). The present study confirmed the possibility of the uptake and translocation of exogenous iodosalicylates in dandelion plants. The highest efficiency of improving iodine content in dandelion was noted after the application of KI at a dose of 50 µM I. This stays in agreement with some of the previous reports regarding plant cultivation in pot experiments [28,37,38]. The most recognized reason for the lower uptake rate noted for IO_3_^−^, as compared to I^−^, is the required reduction of iodates prior to entrance into the plants cell [8]. The second most effective compound, in terms of increasing iodine content in dandelion plants, was 5-iodosalicylic acid. Also, Grzanka et al. [19] reported the higher efficiency of potassium iodide, followed by 5-ISA, in improving iodine levels in young corn plants. The iodine level in both leaves and roots of dandelion plants after the application of 5-ISA was higher than that after using inorganic KIO_3_, particularly at the increased dose (50 µM). Similar observations were reported by Smoleń et al. [18,35] and Sularz et al. [36]. Interestingly, the method of plant cultivation (hydroponic, pot experiment), as well as the type of substrate (peat substrate, mineral soil), had no effect on the observed relationships in lettuce plants [39]. Smoleń et al. [35] has found that the application of 5-ISA at a dose of 8.0 mM I allowed them to achieve a similar effect of iodine biofortification in lettuce leaves as KIO_3_ at a dose of 40.0 mM I. The results of the present study have also allowed us to compare the efficiency of improving the iodine status of dandelion between two iodosalicylates, i.e., 5-ISA and 3,5-diISA. For low iodine doses, there were no differences regarding iodine content in dandelion plants, while the application of 50 µM 5-ISA contributed to a substantial increase in iodine accumulation in leaves and roots, as compared to 3,5-diISA. Interestingly, the level of iodine accumulation after the application of a higher dose of 3,5-diISA was similar to the effect obtained in combination with KIO_3_. That observation stays in agreement with the results presented by Smoleń et al. [39] and Sularz et al. [36]. It is worth mentioning that, apart from iodosalicylates, there also reports evaluating the applicability of other organoiodine compounds, such as iodobenzoates, in plant cultivation [34,40]. Interestingly, in the studies with corn cultivation, the effect of iodine biofortification after the application of 2-IBeA was even similar to that obtained for KI; however, the applied doses were relatively low (10 µM I [19]). 

The results of 5-ISA and 3,5-diISA analysis in the leaves and roots confirmed the possibility of the uptake of iodosalicylates by dandelion plants and indicated its transport within the plant. A significant observation was that 5-ISA and 3,5-diISA accumulated more easily in roots, rather than leaves. The concentration of 5-ISA in plant leaves was approximately two-fold lower than that in roots (irrespective of the dose), while for 3,5-diISA, that ratio was approximately 1:9. On one hand, this confirms the limited but still occurring distribution of intact iodosalicylic acids within the plant. On the other, it suggests the lower mobility of 3,5-diISA in the plant.

Total iodine content was found to be higher in the combination with 5-ISA than with 3,5-diISA. However, the content of 5-ISA in plants grown in the presence of 5-ISA was lower than the content of 3,5-diISA in plants after 3,5-diISA application. This may suggest that 5-ISA, more readily than 3,5-diISA, entered and was involved in biochemical pathways resulting in its transformation into other, most likely inorganic, forms.

The content of 5-ISA in dandelion roots increased not only for the combination with 5-ISA but also in plants grown in the presence of 3,5-diISA (10 and 50 µM I), which suggests the efficient transformation of 3,5-diISA into 5-ISA in dandelion roots. However, the 5-ISA content in the leaves of plants from these combinations was not modified. On the other hand, the increased content of 3,5-diISA was noted in the roots of plants grown in the substrate enriched with 50 µM I of 5-ISA. Smoleń et al. [18] analyzed the content of iodosalicylates in the leaves, roots and root secretions of lettuce plants grown in a hydroponic system. In that study, the presence of 5-ISA was found in plants from the combinations with 5-ISA and 3,5-diISA, and the increased level of 3,5-diISA in roots was noted after the application of 5-ISA. These observations may indicate the occurrence of reverse directions of iodosalicylate transformation within a plant and requires further studies.

The total content of iodine in the control plants was lower than that after exogenous iodine was supplied. However, in both the leaves and roots of dandelion plants from that combination, the presence of low levels of 5-ISA and 3,5-diISA has been detected, which indicated that these iodosalicylates are endogenously synthesized in these plants. These results confirm previous reports on the synthesis of iodosalicylates in higher plants, without exogenous iodine application, as demonstrated for tomato [17], lettuce [18] and sweet corn [19].

One of the most important aspects of widening the popularity and applicability of biofortification approaches in agricultural practice is the maintenance or improvement of the nutritional quality of crop plants. This is also required for the works on herbal and medicinal plants, such as dandelion. One of the most studied parameters is the functioning of the antioxidant system of a plant. On one hand, it may indicate whether the introduced compound, such as iodine or iodosalicylates, acts as a stress factor; on the other hand, the increased content of respective secondary metabolites may be beneficial for the consumers. The study on the evolution of iodine functioning in marine organisms has already suggested it to be one of the first inorganic and efficient antioxidant compounds [41]. Some research has indicated that low iodine doses may also improve the tolerance of higher plants to selected stress factors, such as salinity [42] or heavy metals (Cd, [43]). Blasco et al. [44,45] widely described that the application of inorganic iodine (KI, KIO_3_) during plant cultivation modifies the functioning of the antioxidant system on numerous levels. Numerous studies have revealed, however, that the obtained effects are dose-dependent [45,46,47]. The effect of organic iodine compounds on the antioxidant systems of higher plants has been studied to a lesser extent. Halka et al. [17,40,48] analyzed the changes in the accumulation of respective compounds (ascorbic acid, phenolic compounds), as well as the activity of antioxidant enzymes, under the influence of various iodosalicylates and iodobenzoates during the early growth of tomato plants. Sularz et al. [49] revealed that the application of 10 µM 5-ISA during lettuce cultivation had no significant effect on the contents of total phenols, as well as the antioxidant capacity of lettuce extracts, as measured using DPPH, ABTS and FRAP methods. In other work, similarly low 5-ISA doses (8 µM) did not modify the contents of phenolic compounds in lettuce leaves; however, the content of ascorbic acid—an important antioxidant molecule—decreased [18]. In the present study, two doses of iodine compounds were applied, and the contents of phenolic compounds, SA and JA, as well as the antioxidant capacity, were measured for both dandelion leaves and roots. Greater differentiation of the total contents of phenolic compounds under the influence of the type of iodine compound and doses was noted in dandelion roots, as compared to leaves. The highest root content of phenolics was noted for the combinations with KI application, which was followed by the highest values of antioxidant capacities, as measured using CUPRAC, FRAP and DPPH methods. Importantly, even though the increase in the KI dose from 10 to 50 µM caused almost a 7-fold increase in the iodine content in the roots, the difference with respect to phenolic contents and antioxidant capacity was relatively small. A significant increase in the phenolic content and antioxidant capacity of roots was also noted in the case of a higher dose (50 µM) of 3,5-diISA, even though the iodine level in that combination was lower than that after KI application. As for leaves, the highest content of phenolics was noted for the combinations with 50 µM KIO_3_, but this was not clearly related to the observed values of the antioxidant capacity. The highest antioxidant values, as measured using CUPRAC and FRAP methods, were noted in the leaves of plants treated with 3,5-diISA; however, the content of phenolics was comparable to the control value. That observation suggests that 3,5-diISA may have stimulated the synthesis of other antioxidative compounds not analyzed in the present studies. An interesting observation was a decrease in the SA content in the leaves and roots of plants with almost all combinations of iodine applications. This is consistent with the observation also noted by Halka et al. [40] and Ledwożyw-Smoleń et al. [50] in the studies on young tomato plants. In other work [51], it has been hypothesized that exogenous iodine may modulate the activity of the SAMT gene encoding S-adenosyl-L-methionine:salicylic acid carboxyl methyltransferase, which is responsible for the conversion of SA to its methyl, highly volatile form. Jasmonic acid and its derivatives are plant-signaling molecules classified into a group of phytohormones. Its level can significantly increase under stress conditions caused by biotic and abiotic factors [52,53]. In the present study, the content of jasmonic acid in dandelion leaves increased with all combinations of iodine applications, with the highest values noted for 50 µM 5-ISA and both doses of 3,5-diISA. Conversely, the root level of JA increased only after KIO_3_ treatment and with all remaining combinations, including those with iodosalicylate applications, which were lower than those in the control. Unfortunately, so far, there are no available reports that would provide some further details regarding the possible interaction between iodine and jasmonic acid synthesis.

## 4. Materials and Methods

### 4.1. Reagents and Chemicals Used

The standard reagents, 5-iodosalicylic acid (5-ISA); 3,5-diiodosalicylic acid (3,5-diISA); tetramethylammonium hydroxide (TMAH); Folin–Ciocalteu reagent; chlorogenic acid; 2,2-diphenyl-1-picrylhydrazyl (DPPH); 6-hydroxy-2,5,7,8-tetramethylchroman-2-carboxylic acid (Trolox); neocuproine; ammonium acetate (NH_4_Ac); 2,4,6-Tris(2-pyridyl)-s-triazine were purchased from Sigma-Aldrich Co. LLC, St. Louis, MO, USA. Potassium iodide (KI); potassium iodate (KIO_3_), methanol (99.8%), ethanol (99.8%), were purchased from Chempur, Piekary Śląskie, Poland. Formic acid for LC-MS (97.5–98.5%), gradient-grade acetonitryl for LC, and deuterium-labeled salicylic acid (100 μg mL^−1^ SA-d4 in acetonitrile, Cerilliant^®^) were purchased from Sigma-Aldrich Co., LLC, St. Louis, MO, USA. Ultra-pure water was prepared with the use of a GFL 2304 distiller (Burgwede, Germany) and Hydrolab HLP10 demineralizer (Straszyn, Poland).

### 4.2. Experimental Design and Plant Cultivation

Dandelion plants were cultivated in the spring season of 2020 in a pot experiment located in an Experimental Greenhouse of the Faculty of Biotechnology and Horticulture, University of Agriculture in Krakow (N 50°4′58.1448″, E 19°57′4.0392″). During plant cultivation iodine compounds were applied through fertigation in the following combinations: 1. Control—without iodine; 2—KI, 10 µM I; 3—KI, 50 µM I; 4—KIO_3_, 10 µM I; 5—KIO_3_, 50 µM I; 6—5-iodosalicylic acid (5-ISA), 10 µM I; 7—5-ISA, 50 µM I; 8—3,5-diiodosalicylic acid (3,5-diISA), 10 µM I; 9—3,5-diISA, 50 µM I. The doses of iodine were chosen basing on our own previous studies [38]. Each combination consisted of 4 repetitions with 5 plants per repetition (20 plants per combination). In total, 180 plants were grown in the experiment.

Dandelion (*Taraxacum officinale* F.H. Wiggers coll.) seeds were obtained from a local seed producer. On 30 March 2020, seeds were sown into the cultivation pots (1.5 L capacity) filled with peat substrate mixed with sand at a 2:1 volume ratio, with the characteristics presented in Table 7. The first application of iodine compounds was performed on 18 May 2020 with a volume of 100 mL of respective iodine solutions (or water for the control combination) per each pot. In total, nine applications of iodine compounds (or water) were performed every 5 days during dandelion cultivation. Four days after the last application (30 June 2020), and before reaching the flowering stage, dandelion plants were harvested. The rosettes and roots of plants were collected separately, washed in distilled water, weighted and stored until further analyses.

### 4.3. Sample Preparation

A portion of fresh leaves and roots from each combination was dried at 105 °C for 24 h in order to determine the content of dry matter. The remaining plant material was lyophilized for 48 h with the use of a Christ Alpha 1–4 freeze drier (Martin Christ Gefriertrocknungsanlagen GmbH, Osterode am Harz, Germany), ground in a laboratory grinder, sieved at 0.5 mm (FRITSCH Pulverisette 14, Idar-Oberstein, Germany) and stored in tightly closed polyethylene bags until further analysis.

### 4.4. Total Iodine Analysis

Total iodine content was analyzed in the lyophilized leaves and roots of dandelion plants with the use of an iCAP TQ ICP-MS + 250&1000 Gas Kits mass spectrometer (Thermo Fisher Scientific, Walthom, MA, USA) after alkaline extraction with tetramethylammonium hydroxide (TMAH; [36]). An amount of 0.1 g of lyophilized material (dandelion leaves or roots) was put into 30 mL Falcon tubes, and 10 mL of double-distilled water and 1 mL of 25% TMAH were added and mixed and incubated for 3 h at 90 °C. After cooling, samples were filled up to 30 mL with double-distilled water, mixed and centrifuged for 15 min at 4500 rpm. The supernatants were subjected to the analysis using an ICP-MS/MS spectrometer. I127 was measured in the S-SQ-KED (comprehensive interference removal with helium gas and kinetic energy discrimination) mode with Te (125Te) as an internal standard. The spectrometer was equipped with a quartz cyclonic spray chamber cooled at 2.7 °C, with a MicroMist DC Nebulizer, Ni cones and high-sensitivity interface. The operating plasma power was 1548.6 W, the nebulizer gas flow was 1.092 L/min and the helium flow in the collision cell was 4.3 L/min.

### 4.5. Analysis of Salicylic, Iodosalicylic and Jasmonic Acids

The contents of salicylic (SA), 5-iodosalicylic (5-SA), 3,5-diiodosalicylic (3,5-diISA) and jasmonic acids (JA) were analyzed using an LC-MS/MS system (Ultimate 3000, Thermo Scientific, Waltham, MA, USA, 4500 Q-TRAP Sciex) according to the procedure described in [36]. Here, 50 mg samples of plant material (dandelion leaves and roots) were put into 7 mL polypropylene tubes, and 5 mL of 76% ethanol with 50 ng mL^−1^ deuterium-labeled salicylic acid (SA-d4, Sigma-Aldrich, used as an internal standard) was added, mixed and subjected to ultrasound-assisted extraction for 1 h at 50 °C. After extraction, samples were centrifuged for 5 min at 4 500 rpm. Supernatants were filtered through 0.22 µM nylon syringe filters (FilterBio NY Syringe Filter) and subjected to the analysis using LC-MS/MS. Chromatographic separation was performed using a Luna 3 µm Phenyl-Hexyl 100 Å (150 mm × 3 mm, i.d. 3 µm) column (Phenomenex, Torrance, CA, USA); the following mobile phases were applied (*v*:*v*): A—water + 0.3% formic acid; B—acetonitrile + 0.3% formic acid. The initial volume proportion of the mobile phases (60% A: 40% B) was kept for 2 min and during a further 8 min, increased linearly to reach a 98% B:2% A ratio that was kept for 4 min. After the separation, the initial mobile phase proportion (60% A: 40% B) was restored for 3 min. The injection volume was 10 µL. The mobile phase was directed to an MS ion source between 1 and 14 min of the separation. For detection, electrospray ionization (ESI) in negative ion mode was used for detection, and tandem mass spectrometry MS/MS was used for quantitative studies. Values for the precursor/product transitions of analytes are presented in Appendix A; calibration curves and standard peaks for 5-ISA and 3,5-diISA are shown in Appendix A, while representative chromatographs for iodosalicylates in dandelion leaves and roots are depicted in Appendix A. The Analyst^®^ 1.7 with HotFix 3 software was used for LC-MS/MS control and data processing.

### 4.6. Analysis of Total Phenolic Compounds

The contents of phenolic compounds, as well as the antioxidant potential of dandelion leaves and roots, were determined for 70% methanolic plant extracts. The contents of phenolic compounds in dandelion leaves and roots were analyzed after the reaction with the Folin-Ciocalteu reagent [54]. The leaf or root methanolic extract (0.25 mL) was mixed with 0.25 mL of 25% Na_2_CO_3_, 0.125 mL of the twice-diluted Folin–Ciocalteu reagent (Sigma-Aldrich, diluted twice with water prior to the analysis) and 2.25 mL of water and mixed. After a 15 min incubation in a dark place, the absorbance was measured at 760 nm (JASCO V-530 UV/Vis spectrophotometer). The final results were expressed as mg of chlorogenic acid (Sigma-Aldrich) per 1 g dry weight (chlorogenic acid equivalents, mg CAE g^−1^).

### 4.7. Analysis of Antioxidant Capacity

The radical-scavenging activity (RSC) of dandelion leaves and roots was measured using a colorimetric method monitoring the reduction of the synthetic, stable free radical DPPH (2,2-diphenyl-1-picrylhydrazyl) [55] at 517 nm. A volume of 2.8 mL of 0.1 mM DPPH (Sigma-Aldrich) solution in 96% ethanol was mixed with 0.2 mL of the plant methanolic extract. The DPPH absorbance was measured initially and after 5 min. The RSC results were expressed as μmol Trolox (6-hydroxy-2,5,7,8-tetramethylchroman-2-carboxylic acid, Sigma–Aldrich) per 1 g of dry matter (Trolox equivalents, μmol TE g^−1^).

The antioxidant capacity of dandelion leaves and roots was analyzed by performing CUPRAC (cupric ion-reducing antioxidant capacity; [56]) and FRAP (ferric-reducing antioxidant power; [57]) assays using Trolox (6-hydroxy-2,5,7,8-tetramethylchroman-2-carboxylic acid; Sigma–Aldrich) as an antioxidant standard. The CUPRAC method is based on the measurement of copper (II)-neocuproine reduction by the antioxidants present in the analyzed sample. Volumes of 1 mL of 10 mM CuCl_2_, 1 mL of 7.5 mM neocuproine (Sigma–Aldrich) in 96% ethanol and 1 mL of 1 M NH_4_Ac buffer, pH 7.0, were mixed with 0.3 mL of the methanolic plant extract and 0.8 mL of water. Samples were then incubated for 30 min at room temperature, and then, the absorbance was measured at 450 nm. The results were expressed as μmol Trolox per 1 g of dry matter (Trolox equivalents, μmol TE g^−1^).

The FRAP method measures the rate of the reduction of the ferric–2,4,6-Tris(2-pyridyl)-s-triazine (Fe^3+^–TPTZ) complex by the antioxidants present in the sample [57]. Prior to the analysis, the FRAP working solution was prepared by mixing 300 mM acetate buffer (pH 3.6), 10 mM TPTZ (Sigma-Aldrich) in 96% ethanol and 20 mM FeCl_3_ (10:1:1, *v*:*v*:*v*). Then, 3 mL of FRAP working solution were mixed with 0.1 mL of the plant methanolic extract and 0.3 mL of water and incubated for 30 min at room temperature. Further on, the absorbance was measured at 595 nm. The results were expressed as μmol Trolox per 1 g of dry matter (Trolox equivalents, μmol TE g^−1^).

### 4.8. Statistical Analysis

Statistical verification of the obtained data was conducted with the use of the ANOVA module of Statistica 13.3 software at a significance level of <0.05.

## 5. Conclusions

Plant biofortification with iodine has been proposed as one of the approaches for improving the iodine status of the human population. It seems to be of great importance particularly for those who consume a mostly plant-based diet. One of the novel directions in these studies include the usage of organoiodine compounds as iodine sources for plants, as well as evaluating the possibility of increasing the iodine content in herbal and medicinal plants. In the current study, the efficiency of the uptake, accumulation and distribution of iodine by dandelion plants was evaluated. It was confirmed that iodosalicylates (5-iodosalicylic acid and 3,5-diiodosalicylic acid) are taken up and translocated within a plant. The application of 5-iodosalicylic acid during dandelion cultivation allowed to obtain the high accumulation of iodine in both the leaves and roots, even exceeding the level noted for the commonly applied inorganic iodine source, i.e., KIO_3_. It was demonstrated that after root uptake, iodosalicylates are, to a limited extent, transported into the leaves. The analysis of the plant biomass and antioxidant capacity of leaf and root extracts allows us to conclude that the applied iodine compounds and doses did not exhibit toxic effects on dandelion plants. Based on the obtained results, future studies are needed in order to characterize the metabolic processes that iodosalicylates undergo prior to or after entrance into a root cell. The interaction between iodosalicylates and the synthesis of various secondary metabolites, including those with antioxidant properties, are also yet to be discovered, particularly as they constitute the health-promoting quality of medicinal plants.

## Figures and Tables

**Table 1 molecules-28-05638-t001:** Plant biomass and dry matter content of dandelion plants.

Combination	Leaf Weight per Plant (g)	Root Weight per Plant (g)	Total Weight of a Plant (g)	% d.m. Leaves	% d.m. Roots
Control	20.3 ± 0.87 a ^1^	37.9 ± 12.33 a	58.2 ± 12.03 a	14.3 ± 0.02 c	21.6 ± 0.24 cd
KIO_3_ 10 µM	18.8 ± 0.95 a	48.8 ± 3.47 a	67.5 ± 4.11 a	13.1 ± 0.03 a	22.8 ± 0.27 d
KIO_3_ 50 µM	16.3 ± 1.03 a	46.3 ± 2.95 a	62.5 ± 3.86 a	13.1 ± 0.06 a	23.3 ± 0.23 d
KI 10 µM	19.5 ± 0.65 a	54.0 ± 9.06 a	73.5 ± 9.18 a	13.7 ± 0.04 b	19.8 ± 0.44 bc
KI 50 µM	19.3 ± 1.84 a	70.8 ± 18.60 a	90.0 ± 20.12 a	15.1 ± 0.09 e	15.9 ± 0.54 a
5-ISA 10 µM	19.0 ± 1.78 a	70.3 ± 21.12 a	89.3 ± 22.00 a	14.3 ± 0.06 c	22.0 ± 0.25 cd
5-ISA 50 µM	19.3 ± 1.55 a	60.5 ± 12.81 a	79.8 ± 13.60 a	13.5 ± 0.04 b	18.4 ± 0.75 b
3,5-diISA 10 µM	22.5 ± 1.85 a	77.5 ± 18.46 a	100.0 ± 19.79 a	14.7 ± 0.06 d	20.4 ± 0.81 bc
3,5-diISA 50 µM	20.0 ± 1.47 a	65.5 ± 13.74 a	85.5 ± 14.09 a	14.3 ± 0.04 c	19.1 ± 0.47 b

^1^ Data (mean ± SE) in columns followed by the same letter are not significantly different at *p* < 0.05 (*n* = 4).

**Table 2 molecules-28-05638-t002:** The contents of total iodine, 5-iodosalicylic acid (5-ISA) and 3,5-diiodosalicylic acid (3,5-diISA) in dandelion leaves and roots.

Combination		Leaves			Roots	
	Iodine (mg I kg^−1^)	5-ISA (ng g^−1^)	3.5-diISA (ng g^−1^)	Iodine (mg I kg^−1^)	5-ISA (ng g^−1^)	3.5-diISA (ng g^−1^)
Control	0.3 ± 0.01 a ^1^	0.32 ± 0.03 a	0.20 ± 0.67 a	0.2 ± 0.01 a	0.1. ± 0.03 a	0.7 ± 0.20 a
KIO_3_ 10 µM	7.8 ± 0.08 b	0.08 ± 0.02 a	0.24 ± 0.09 a	1.7 ± 0.02 b	0.6 ± 0.14 abc	1.2 ± 0.26 a
KIO_3_ 50 µM	49.6 ± 0.77 e	0.12 ± 0.06 a	0.56 ± 0.22 a	9.1 ± 0.22 d	0.3 ± 0.03 ab	1.1 ± 0.10 a
KI 10 µM	13.4 ± 0.47 c	0.31 ± 0.05 a	0.31 ± 0.08 a	2.8 ± 0.04 c	0.1 ± 0.02 a	0.9 ± 0.15 a
KI 50 µM	133.5 ± 1.89 g	0.13 ± 0.05 a	0.27 ± 0.08 a	19.1 ± 0.15 f	0.8 ± 0.17 bc	0.3 ± 0.05 a
5-ISA 10 µM	9.2 ± 0.12 bc	0.46 ± 0.14 a	0.38 ± 0.16 a	2.3 ± 0.01 bc	0.9 ± 0.08 cd	0.3 ± 0.05 a
5-ISA 50 µM	99.2 ± 1.74 f	3.84 ± 0.21 b	0.34 ± 0.12 a	17.1 ± 0.44 e	6.3 ± 0.18 e	21.7 ± 0.54 b
3,5-diISA 10 µM	7.2 ± 0.12 b	0.46 ± 0.12 a	2.29 ± 0.23 b	2.9 ± 0.02 c	1.0 ± 0.05 cd	22.2 ± 0.43 b
3,5-diISA 50 µM	38.5 ± 0.31 d	0.40 ± 0.03 a	8.88 ± 0.54 c	9.1 ± 0.13 d	1.4 ± 0.19 d	95.8 ± 1.51 c

^1^ Data (mean ± SE) in columns followed by the same letter are not significantly different at *p* < 0.05 (*n* = 4).

**Table 3 molecules-28-05638-t003:** The contents of phenolic compounds, salicylic acid (SA) and jasmonic acid (JA) in dandelion leaves.

Combination	Phenolic Compounds (mg CAE g^−1^)	SA Content (ng g^−1^)	JA Content (ng g^−1^)
Control	41.9 ± 0.48 bc ^1^	58.6 ± 1.56 c	83.2 ± 9.35 a
KIO_3_ 10 µM	43.5 ± 0.54 c	41.1 ± 1.41 ab	122.6 ± 7.36 b
KIO_3_ 50 µM	50.2 ± 1.11 d	47.3 ± 1.03 b	120.4 ± 1.28 b
KI 10 µM	37.2 ± 0.423 a	43.6 ± 0.97 ab	156.5 ± 4.78 cd
KI 50 µM	39.5 ± 1.05 abc	69.0 ± 1.44 d	132.2 ± 2.54 bc
5-ISA 10 µM	43.7 ± 0.14 c	41.5 ± 1.70 ab	138.3 ± 6.48 bc
5-ISA 50 µM	36.5 ± 1.45 a	38.4 ± 0.62 a	164.6 ± 3.26 de
3,5-diISA 10 µM	38.8 ± 0.59 ab	45.2 ± 1.89 b	182.4 ± 4.25 e
3,5-diISA 50 µM	39.8 ± 1.29 abc	42.2 ± 0.80 ab	175.2 ± 2.92 de

^1^ Data (mean ± SE) in columns followed by the same letter are not significantly different at *p* < 0.05 (*n* = 4).

**Table 4 molecules-28-05638-t004:** The contents of phenolic compounds, salicylic acid (SA) and jasmonic acid (JA) in dandelion roots.

Combination	Phenolic Compounds (mg CAE g^−1^)	SA Content (ng g^−1^)	JA Content (ng g^−1^)
Control	8.11 ± 0.10 a ^1^	6.0 ± 0.14 d	383.1 ± 3.63 f
KIO_3_ 10 µM	8.0 ± 0.17 a	2.5 ± 0.12 a	432.0 ± 1.42 g
KIO_3_ 50 µM	8.6 ± 0.19 ab	4.3 ± 0.23 bc	439.3 ± 3.44 g
KI 10 µM	10.2 ± 0.13 d	4.5 ± 0.26 c	112.1 ± 3.55 a
KI 50 µM	11.7 ± 0.12 e	3.6 ± 0.11 b	157.5 ± 1.64 b
5-ISA 10 µM	8.6 ± 0.06 ab	5.1 ± 0.30 c	258.7 ± 10.60 e
5-ISA 50 µM	9.0 ± 0.11 bc	5.2 ± 0.22 cd	190.9 ± 0.49 c
3,5-diISA 10 µM	8.14 ± 0.13 a	4.6 ± 0.03 c	218.4 ± 3.11 d
3,5-diISA 50 µM	9.7 ± 0.16 cd	2.4 ± 0.15 a	166.9 ± 2.57 b

^1^ Data (mean ± SE) in columns followed by the same letter are not significantly different at *p* < 0.05 (*n* = 4).

**Table 5 molecules-28-05638-t005:** Antioxidant capacity of the extracts of dandelion leaves.

Combination	Antioxidant Capacity (µmol TE g^−1^)		
	CUPRAC	FRAP	DPPH
Control	187.1 ± 2.66 a ^1^	150.6 ± 3.25 a	67.8 ± 2.09 a
KIO_3_ 10 µM	201.4 ± 6.09 abc	153.9 ± 2.15 ab	71.1 ± 1.32 ab
KIO_3_ 50 µM	204.5 ± 4.09 abc	168.4 ± 3.08 bc	72.8 ± 1.88 b
KI 10 µM	209.2 ± 2.89 bc	173.7 ± 1.92 c	69.7 ± 0.12 ab
KI 50 µM	194.8 ± 2.74 ab	164.6 ± 4.48 abc	70.0 ± 0.90 ab
5-ISA 10 µM	201.2 ± 1.73 abc	171.6 ± 5.10 c	71.2 ± 1.00 ab
5-ISA 50 µM	197.0 ± 4.21 abc	170.8 ± 1.21 c	69.3 ± 0.66 ab
3,5-diISA 10 µM	214.6 ± 4.25 c	178.8 ± 4.23 c	67.2 ± 1.62 ab
3,5-diISA 50 µM	215.1 ± 4.66 c	180.8 ± 3.40 c	66.1 ± 1.14 a

^1^ Data (mean ± SE) in columns followed by the same letter are not significantly different at *p* < 0.05 (*n* = 4).

**Table 6 molecules-28-05638-t006:** Antioxidant capacity of the extracts of dandelion roots.

Combination	Antioxidant Capacity (µmol TE g−1)		
	CUPRAC	FRAP	DPPH
Control	68.8 ±3.06 abc ^1^	24.4 ± 0.27 ab	23.7 ± 0.69 ab
KIO_3_ 10 µM	66.9 ± 1.77 ab	23.9 ± 0.44 a	25.3 ± 0.82 abc
KIO_3_ 50 µM	76.4 ± 1.45 cd	26.8 ± 0.29 cd	26.6 ± 0.61 bc
KI 10 µM	84.0 ± 0.93 d	31.9 ± 0.57 e	30.9 ± 0.27 d
KI 50 µM	85.2 ± 2.11 d	34.8 ± 0.40 f	28.7 ± 0.65 cd
5-ISA 10 µM	71.1 ± 1.38 bc	26.1 ± 0.46 bc	24.9 ± 0.63 ab
5-ISA 50 µM	67.3 ± 1.78 ab	25.8 ± 0.60 abc	23.7 ± 0.79 ab
3,5-diISA 10 µM	61.8 ± 1.11 a	24.1 ± 0.45 ab	22.5 ± 0.37 a
3,5-diISA 50 µM	72.2 ± 2.54 bc	28.8 ± 0.50 d	27.2 ± 1.54 bcd

^1^ Data (mean ± SE) in columns followed by the same letter are not significantly different at *p* < 0.05 (*n* = 4).

**Table 7 molecules-28-05638-t007:** Main physicochemical characteristics of the substrate used in the pot experiment with dandelion cultivation.

pH	EC	Eh	N-NH_4_^+^	N-NO_3_^−^	Ca	K	Mg	Na	P	S	Cl^−^	I
	**mS/cm**	**mV**	**mg dm^−3^**									**mg·kg^−1^**
4.01	0.26	383.8	96.8	83.7	1913.3	196.2	72.6	20.9	12.3	1705.3	13.6	2.0

## Data Availability

All data are available via email request to the corresponding author.

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
