# Peer review of "Iodine Biofortification of Dandelion Plants (Taraxacum officinale F.H. Wiggers Coll.) with the Use of Inorganic and Organic Iodine Compounds"

_molecules, 2023, doi:10.3390/molecules28155638_

Round 1

Reviewer 1 Report

1. Line 60. “…through chloride channel ions” were the ion channels meant here?

2. Why iodosalicylates were chosen for organic source of iodine? Some brief explanation in the introduction section would be useful.

3. How was the concertation for the experiments chosen? Why exactly 10 and 50 µM?

4. Why were SA and JA selected to track in the plant tissues? Such choice is not evident.

5. Line 243. What is 2-IBeA?

6. There is no data regarding the ICP-MS/MS experiment.

7. Line 407. “…according to the procedure described in [32]”. There is no information regarding LC-MS/MS method used for the analysis. Moreover, the cited paper claims that fatty acid profile was studied by LC-MS/MS, while further information regarding the analytical setup describes GC-MS method. The authors should provide the experimental conditions used for analysis of 5-ISA and 3,5-diISA. As soon as the detailed procedure would be provided, further discussion may appear.

 8. The last statement in the abstract wasn’t revealed in the main text: “The chemical analyses have revealed that iodosalicylates are endogenous compounds of dandelion plants”. It is not clear how authors came to such conclusion. Were the isotope labeled standards used? This conclusion needs to be explained.

Some minor revision required in terms of English. The text is well structured, but some phrases should be reconsidered. 

Reviewer 2 Report

Manuscript: „Iodine biofortification of dandelion plants (Taraxacum officinale F.H. Wiggers coll.) with the use of inorganic and organic iodine compounds“

Iwona Ledwożyw-Smoleń, Joanna Pitala, Sylwester Smoleń, Marta Liszka-Skoczylas, Peter Kováčik

This research covers many tasks associated with agriculture and biotechnological approach. The authors evaluated the efficiency of the uptake and accumulation of iodine in the plants of dandelion grown in pot experiment. Dandelion plants were cultivated in an Experimental Greenhouse of the Faculty of Biotechnology and Horticulture, University of Agriculture in Krakow. Iodine was applied through fertigation using inorganic (KI, KIO3) and organic forms (5-iodosalicylic acid, ISA; 3,5-diiodosalicylic acid, 3,5-diISA) in two concentrations (10 and 50 µM). The content of total iodine and iodosalicylic acids as well the plant biomass and antioxidant capacity of dandelion leaves and roots were analyzed. The highest efficiency of improving iodine content in dandelion leaves and roots was noted for 50 µM KI. These findings could be valuable to explore the influence of iodine compounds on secondary metabolism of medicinal plants.

Comments to the authors:

1)      It should be correct S-adenosyl-L-methionine:salicylic acid carboxyl methyltransferase;

2)     It should be correct 3 h at 90 °C in the paragraph 402;

3)     It should be correct: 2,4,6-Tris(2-pyridyl)-s-triazine in the paragraph 441;

4)      It should be correct Fe3+ in paragraph 442.

Reviewer 3 Report

The manuscript entitled "Iodine biofortification of dandelion plants (Taraxacum officinale F.H. Wiggers coll.) with the use of inorganic and organic iodine compounds" and signed by Iwona Ledwożyw-Smoleń et al. presents findings regarding the iodine biofortification of dandelion plants with both inorganic forms and organic forms. The characterization of the biofortified plants at the minimum required and simple, but well done. Despite its medium originality (organic iodine compounds was previously used for biofortification, the conclusion should be corrected), the topic and the findings are important for the scientific community. Some judgements and indications for improvement of the manuscript are presented bellow.

1. Abstract: The context/motivation and the aims are clearly presented. The main results and conclusions are also indicated.

2. Introduction: The health/clinical context of the iodine biofortification is well discussed as well as the biofortification process. Overall, the introduction is well written.

3. Results: The results are well organized, the statistical significance was tested for all analysis tat was done. Some representative chromatograms (at least as supplementary) should be presented so that the readers may judge the quality of the chromatographic analysis. Also, some identification results (MS/MS spectra) for the iodine-containing compounds would be welcome.

3. The Materials and Methods section needs subsections for better clarity and accessibility. This section should start with a subsection dedicated for the reagents and chemicals used and their purity grade and origin of procurement. Other possible sections: Experimental design of plant growing, Sample preparation, Iodine analysis, Antioxidant analysis, etc. The authors cited the chromatographic procedure but some brief data (without many details) should be introduced here too. What method? Was it an external calibration curve or a standard addition method?

4. The notation µM TE  g–1 is not corect. It is rather µmol TE/g since µM refers to µmol/L and it cannot be µmol/L in 1 g sample. 

5. Table 1 appears the last in the manuscript, the Tables should be renumbered. In this Table it is unclear what Eh (mV) means and what is its usefulness. Also, are you sure that this growing substrate contains 2 mg/L? It is a lot. Also, the S content of 1705 mg/dm3 is huge. The authors should check all these data.

Round 2

Reviewer 1 Report

Everything is clear now

Reviewer 3 Report

The authors did answers all questions and the manuscript could be accepted for publication in its current form.